

# Spatiotemporal dynamics of fog and low clouds in the Namib unveiled with ground and space-based observations

Hendrik Andersen[1,2], Jan Cermak[1,2], Irina Solodovnik[1,2], Luca Lelli[3], and Roland Vogt[4]

[1]Karlsruhe Institute of Technology (KIT), Institute of Meteorology and Climate Research
[2]Karlsruhe Institute of Technology (KIT), Institute of Photogrammetry and Remote Sensing
[3]University of Bremen, Institute of Environmental Physics and Remote Sensing
[4]University of Basel, Department of Environmental Sciences

**Correspondence:** Hendrik Andersen (hendrik.andersen@kit.edu)

**Abstract.** Fog is an essential component of Namib-region ecosystems. Current knowledge on Namib-region fog patterns and processes is limited by a lack of coherent observations in space and time. In this study, data from multiple satellite platforms and station measurements paint a coherent picture of the spatiotemporal dynamics of fog and low cloud (FLC) distribution. It is found that observed seasonal patterns derived from satellite observations differ from station measurements in coastal locations,

whereas they agree further inland. This is linked to an observed seasonal cycle in the vertical structure of FLC that determines the probability of low-level clouds touching the ground. For the first time, these observations are complemented by spatially coherent statistics concerning the diurnal cycle of FLC using geostationary satellite data. The average timing of the start of the diurnal FLC cycle is found to strongly depend on the distance to the coastline (r ≈ 0.85 north of 25°N), a clear indication of dominant advective processes. In the central Namib, FLC typically occurs 2–4 hours later than in other coastal regions,

possibly due to local advection patterns. The findings lead to a new conceptual model of the spatiotemporal dynamics of fog and low clouds in the Namib.

*Copyright statement.* TEXT

## 1 Introduction

In arid environments like the Namib, fog can be a crucial source of water for many species and ecosystems (e.g. Seely et al.,

1977; Seely, 1979; Shanyengana, 2002; Ebner et al., 2011; Azúa-Bustos et al., 2011; Roth-Nebelsick et al., 2012; Eckardt et al., 2013; McHugh et al., 2015). However, only little is known about its spatial and temporal patterns, as well as the environmental drivers of fog in the Namib.

    While meteorological measurements are generally sparse in this region, historical station observations of fog in the central Namib between the 1940s and the 1980s have shown contrasting seasonal patterns of fog occurrence at coastal and inland

locations (Nagel, 1959; Lancaster et al., 1984). These studies find that at inland locations, fog tends to occur less frequently between April and August, while fog occurrence at coastal locations peaks during this time. More recently, satellite data have





been used to study the patterns fog and low clouds (FLC) in the Namib (e.g. Olivier, 1995; Cermak, 2012; Andersen and Cermak, 2018). The only satellite-based study that comprises a multi-year seasonal cycle of FLC is presented in Cermak (2012), and while the observed patterns compare well to station measurements presented in (Lancaster et al., 1984) at the inland station in Gobabeb, observed seasonal cycles from satellite data and station measurements differ at the coastal location

in Walvis Bay. This could be related to seasonal cycles of formation mechanisms or in vertical characteristics of FLC in this region, i.e. the fact that all low clouds are treated summarily by the satellite technique, whereas only the ones with the lowest cloud bases manifest themselves as fog as reported by ground-based observations. However, a spatially coherent detailed characterization of FLC, including vertical characteristics, as well as seasonal and diurnal patterns is still missing. Uncertainties also exist related to the mechanisms that lead to fog formation. While most studies (e.g. Lancaster et al., 1984; Olivier and

Stockton, 1989; Olivier, 1995; Cermak, 2012; Andersen and Cermak, 2018) relate Namib-region fog mostly to the advection of low clouds formed over the cool waters of the Benguela current, recent analyses of stable isotopes have pointed to mixed or sweet water sources, which has been interpreted as an indication for radiation fog (Kaseke et al., 2017, 2018). However, the labor-intensive field work needed for isotope analyses has limited these studies in spatial and temporal extent, underscoring the need for a spatiotemporally complete and coherent characterization of FLC mechanisms. In this study, active-sensor and

passive-sensor satellite data are used in conjunction with ground-based meteorological measurements to better understand fog and low-cloud patterns at different scales. The goal of this study is to provide climatological, spatiotemporally complete patterns that help understand the processes driving Namib-region fog and low clouds.

The guiding hypotheses are that

1. FLC patterns in time and space differ distinctly between the coastline and regions further inland.

2. Apparent differences between the seasonal cycle of fog as observed from the ground and satellite perspectives are explained by a seasonal cycle in the vertical structure of FLC.

## 2 Data and methods

In this study, multiple data sets from various space-based sensors are used to characterize FLC and analyze its spatiotemporal occurrence patterns. The general spatial domain investigated in this study is the western coastline of southern Africa (13.5°S–

34°S and 5°E–20°E, Fig. 1 a)), with a specific focus on patterns over land and a core region of FLC occurrence in the central Namib nearby Walvis Bay (22.5°S–24°S and ≈14°E–15.5°E, Fig. 1 b)). In the central Namib, the FogNet station network (Kaspar et al., 2015) is located, providing a ground-based perspective on fog patterns. Detailed descriptions of the different sensors, techniques and data sets used in this study are given in sections 2.1–2.4.

### 2.1 SEVIRI

Coherent spatiotemporal patterns of FLC occurrence are created using data from the Spinning-Enhanced Visible and Infrared Imager (SEVIRI) aboard the Meteosat Second Generation (MSG) satellites. The sensor has a nadir spatial resolution of 3 km





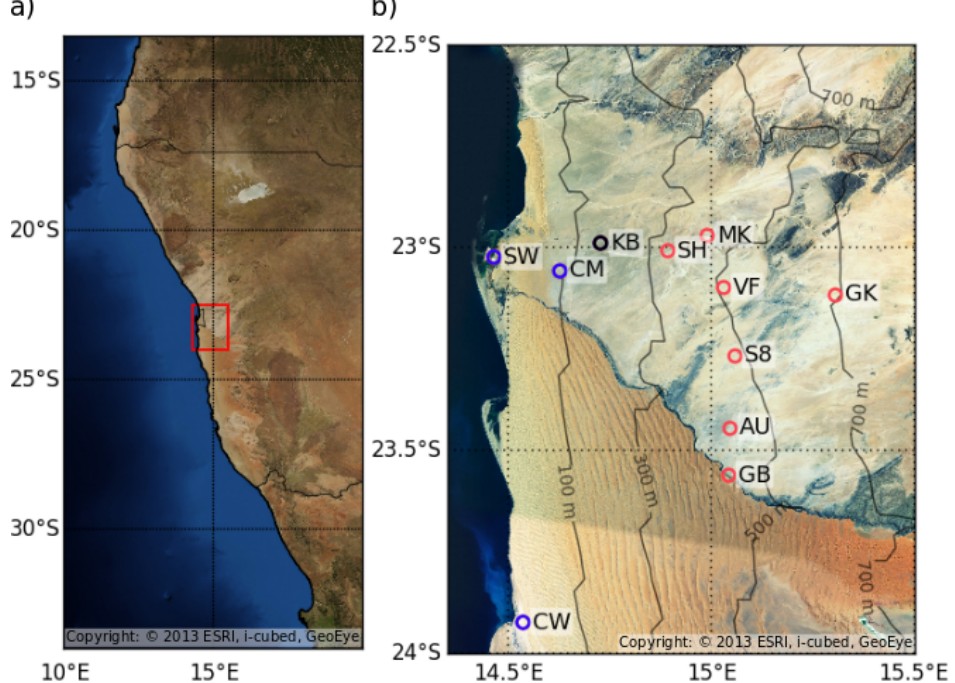

**Figure 1.** a) An overview of the study area. The red box highlights the central Namib, which is shown in more detail in b): FogNet stations are illustrated by circles and are annotated with their respective IDs (full station names are given in the appendix). Blue circles represent coastal, and red circles inland stations as defined in this study (section 2.4). The station Kleinberg (KB) is colored in black, as it is viewed to be at a transitional location not clearly belonging to either category.

and provides 96 hemispheric scans per day (repeat rate of 15 minutes) (Schmetz et al., 2002). The novel FLC-detection technique by Andersen and Cermak (2018) is applied to data of nearly the entire operational period of MSG satellites (2004–2017). The technique uses only observations in the thermal infrared, enabling a fully-diurnal detection of FLC in the region. It has shown good skill in a validation against surface net radiation measurements with a probability of detection of 94 %, a false alarm rate of 12 % and a general correctness of all classifications of 97 % (Andersen and Cermak, 2018).

## 2.2 CALIPSO

SEVIRI observations are complemented by retrieved layer heights from the active-sensor platform from Cloud-Aerosol LiDAR and Pathfinder Satellite Observations (CALIPSO). Mounted onboard the satellite is the Cloud-Aerosol Lidar with Orthogonal Polarization (CALIOP) that samples with 30 m vertical and 333 m horizontal resolutions. Here, the CALIPSO level 2 5 km cloud-layer product (version 4.10) is used to detect FLC with the algorithm developed by Cermak (2018) for the period of June, 13th 2006 – December, 31st 2017 (daytime and night-time). The algorithm essentially detects low clouds with a cloud-top altitude of $<= 2000$ m and a cloud-base altitude $<= 500$ m above ground level. Additionally, spatial and temporal patterns



of cloud-top height (CTH) are generated using the same data. Results are then aggregated to 2.5°x2.5° regions to increase the sample size as in Cermak (2018).

## 2.3 SCIAMACHY

The cloud bottom altitude from the SCanning Imaging Absorption spectroMeter of Atmospheric CHartographY (SCIA-

MACHY) (Bovensmann et al., 1999), on board Envisat, has been inferred from the fit of sunlight absorption by the strongest molecular band of oxygen (the A-band), located in the near-infrared (NIR) between 750-770 nm, at a nominal spectral resolution of 0.4 nm. The deployed algorithm Semi-Analytical CloUd Retrieval Algorithm (SACURA) (Rozanov and Kokhanovsky, 2004; Lelli et al., 2012) exploits the constant vertical abundance of columnar oxygen so that any cloud intervening in the field-of-view of the sensor shields the gas column below, thus changing the depth of the A-band. Concurrently, the increase

of absorption by oxygen within a cloud due to multiple scattering is accounted for by calculating the single-scattering albedo of the atmospheric volume at 760 nm. In this way, with the knowledge of the cloud optical thickness (COT) computed at the non-absorbing channel 758 nm, the inversion of the measurement delivers the cloud geometrical extent. As long as the sensed cloud is single-layered and has a constant liquid water content, the reported model error in CBH amounts to -200/350 m (Lelli et al., 2011), which is paired to a CTH absolute error of $\pm 250$ m (Lelli et al., 2012, 2014), irrespective of COT and given

CTH values $< 10$ km. However, the coarse footprint size of SCIAMACHY (60 x 40 km$^2$ at nadir) can degrade this assumption due to likely heterogeneity of the cloud field sensed by the instrument. In this case, a set of filters ensures the extraction of a representative cloud sample from the unfiltered data record, discarding cirrus and multi-layer clouds. The procedure employed here is extensively described in Lelli and Vountas (2018) and 7 years (2003–2009) of retrievals at the SCIAMACHY overpass local time of $\approx 10$:15 AM are monthly aggregated at a grid resolution of 0.5°.

## 2.4 Ground-based measurements

Three years (2014–2017) of station measurements from the FogNet station network in the central Namib are used to gain insights into fog occurrence at the ground. As illustrated in Fig. 1 b), the FogNet network consists of 11 automated meteorological stations that are aligned in two transects (N-S from 22.97°S–23.92°S and W-E from 14.46°E–15.31°E). FogNet was created as part of the Southern African Science Service Centre for Climate Change and Adaptive Land Management (SASSCAL)

initiative to study fog occurrence and processes in this region (Kaspar et al., 2015). The stations can be broadly classified by their geographic location into low-lying coastal stations (blue: all stations located $< 100$m above sea level (ASL)) and inland stations (red: all stations located $> 300$m ASL), as well as a transition station (Kleinberg: KB).

Measurements of fog precipitation and relative humidity are combined to create a binary data set of fog occurrence. Fog precipitation measurements describe advected cloud water collected by a Juvik fog collector (Juvik and Nullet, 1995). The

Juvik fog collector is an omnidirectional, cylindrical aluminium fog gauge, positioned at 1.5 m above ground. Measured fog precipitation depends on the near-ground liquid water content of fog, fog droplet size, and also scales with near-surface wind speed, as this determines the volume of air that perfuses the gauge (Frumau et al., 2011). There can be a time lag between fog occurrence and measured fog precipitation, due to the build-up time until the runoff of fog water occurs. Also, the instrument



might not be sensitive in instances of very thin fog, as there is a lower limit of water needed for runoff. To reduce measurement-related uncertainties in the fog occurrence estimates, fog precipitation measurements are supplemented by observations of 2 m relative humidity (Campbell CS215) to create a binary fog product. The station measurements have a one-minute temporal resolution but are averaged in 15-minute intervals for comparison with SEVIRI observations. Fog is counted whenever the

average relative humidity during a 15-minute interval exceeds 95 % or any amount of fog precipitation is measured during this time.

A ceilometer (Vaisala CL31, instrument "CL31-2" in Wiegner et al. (2018)) complemented the measurements at the station Coastal Met (CM) from September 2017 to June 2018 to observe patterns in cloud-base height. In July, the ceilometer was repositioned closer to the coastline (Swakopmund). The CL31 emits a laser beam at 905 nm and provides a profile of attenuated

light backscatter with a vertical resolution of up to 5 m (Martucci et al., 2010; Kotthaus et al., 2016). It emits $2^{14}$ laser pulses with a frequency of 10 kHz every 2 seconds, after which it takes about 0.36 seconds of idle time to compute the cloud base height (Vaisala CL31 firmware) (Kotthaus et al., 2016). CBH retrievals are then averaged to a temporal resolution of one minute. This CL31 has a minimum detection altitude of $\approx$ 40 m and was located at $\approx$ 95 m above sea level (ASL) at CM and is currently situated at $\approx$ 19 m ASL at Swakopmund. As such, the ceilometer cannot give an accurate estimate for CBH < 135 m

ASL or 59 m, respectively. Here, data from one year (September 2017 to August 2018) are used. Due to data collection difficulties, no data is available during February 2018. To focus on fog and low-level clouds, only CBH < 2000 m AGL are considered.

## 3  Results and discussion

### 3.1  Fog and low cloud patterns and seasonality

Figure 2 shows climatological patterns of FLC occurrence as seen by CALIPSO (Fig. 2 a)) and SEVIRI (Fig. 2 b)) using the algorithms developed by Cermak (2018) (land and ocean) and Andersen and Cermak (2018) (land only), respectively. The spatial patterns of FLC occurrence correspond well with those derived in earlier satellite-based studies (Olivier, 1995; Cermak, 2012; Andersen and Cermak, 2018), where FLC occurs frequently over the ocean and along the coastline, with three separate core regions over land: the southern parts of the Angolan coastline (15°S–17°S), the Namibian coastline from Walvis

Bay ($\approx$ 23°S) northwards to 18°S, and to a lesser extent at Alexander Bay at the Namibian-South African border ($\approx$ 28°S). The spatial patterns of FLC occurrence (Fig. 2 a) and b)) indicate a connection between the stratocumulus cloud field off the southwestern African coastline and FLC occurrence over , even though the CALIPSO data is not able to capture some of the finer spatial features in FLC distribution (e.g., the low-FLC region between 17°S and 18°S) due to the coarse averaging resolution. The occurrence of FLC along the western coast of southern Africa features a distinct seasonal cycle that varies with

latitude (Fig. 2 c)), and agrees with findings from Cermak (2012). The observed seasonal pattern of the Angolan Namib agrees well with that of the southeastern Atlantic stratocumulus cloud field (Klein and Hartmann, 1993) and underscores a likely link of stratocumulus clouds over the ocean and FLC over land. The latitudinal dependence of the seasonal patterns of FLC may be



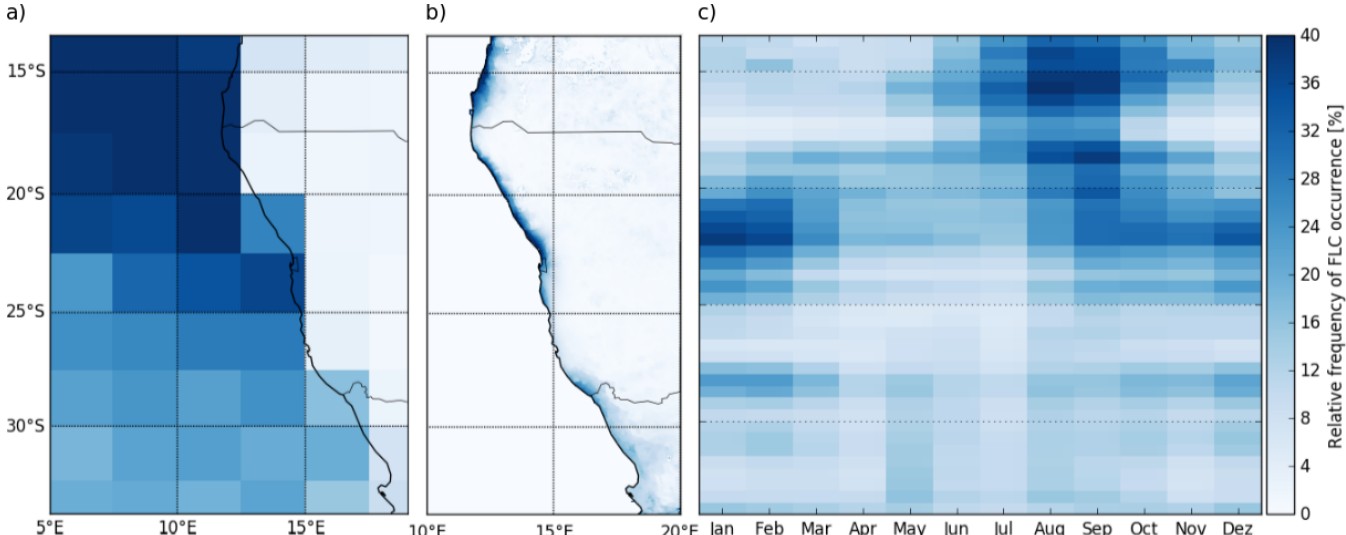

**Figure 2.** A satellite-based climatology of relative FLC occurrence frequency derived by using the algorithms presented in Cermak (2018) (a)) and Andersen and Cermak (2018) (b)), based on the nearly complete data records of CALIPSO (2006–2017) and SEVIRI (2004–2017). The seasonality (c)) is computed by averaging pixels from (b) in coastal regions (maximum 100 km distance to coastline) with frequent FLC occurrence (minimum of 5 % relative FLC occurrence in the 14-year climatology shown in b)).

an indication of a seasonal shift of the dynamical systems responsible for a landward advection of low clouds formed over the ocean.

The satellite-derived seasonal cycle of FLC occurrence agrees well qualitatively and quantitatively with the seasonality of fog observed at inland stations (Fig. 3 b)). However, in accordance to the comparison of results from Lancaster et al. (1984)

and Cermak (2012), the observations do not show similar patterns at the coastal stations (Fig. 3 a)). Here, satellite observations show a seasonality that resembles that found at inland stations, with a minimum during May and a maximum in September. In contrast, ground-based fog observations at the coastal stations peak in winter between April and August. It should be noted that while the seasonal patterns disagree, during the period from April to July, observed fog/FLC occurrence frequencies agree quantitatively (Fig. 3 b)). Both - similarities and discrepancies of the observed seasonal cycles - are likely explained

in large parts by the seasonality in the vertical structure of FLC in the central Namib (Fig. 3 b)). Cloud-vertical properties are investigated using ground-based and space-based active sensoric. A distinct seasonal pattern in cloud-top height (CTH) is observed using CALIPSO, with 183 m lower cloud-top altitudes between April and June compared to the rest of the year (significant at the 99 % confidence level: independent t test). This seasonal pattern is also found in observations of cloud-base height (CBH) of the CL31 ceilometer located in CM. Here, cloud bases are found to be on average 130 m lower between

April and June than during the rest of the year. As the ceilometer measurements are only available for one (incomplete) year, 7-year monthly averaged CBH estimates from SCIAMACHY are considered in addition. While the SCIAMACHY-derived CBH are especially low later in the year (June and July), the seasonal pattern agrees in the sense that it features lower CBH





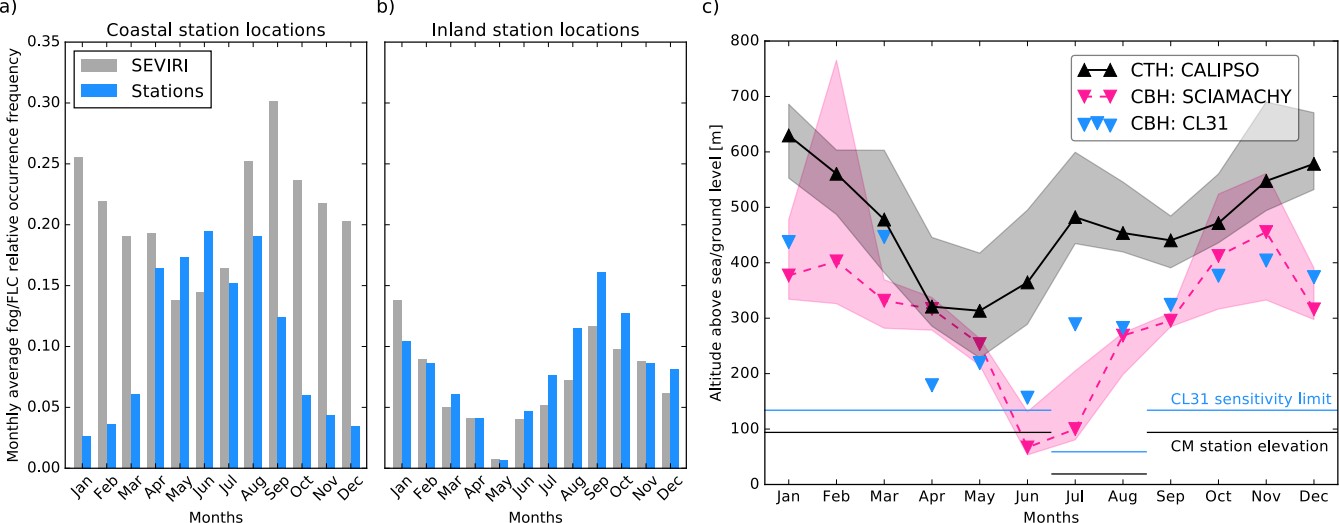

**Figure 3.** Monthly averaged relative fog/FLC occurrence frequency at locations of coastal (a)) and inland stations (b)). SEVIRI observations (2004–2017) are illustrated by grey bars, station measurements of ground fog (2015–2017) in blue. c) Medians, 25th and 75th percentiles of monthly averaged CBH and CTH in the central Namib based on SCIAMACHY (AGL; 22.5°S-24.0°S and 14.25°E–15.5°E, 2003–2009) and CALIPSO (ASL; 22.5°S-24.0°S and 14.0°E–15.5°E, 2006–2017) observations, respectively. CL31 CBH observations (ASL) are only available since September 2017. CL31 station elevation and sensitivity limit are illustrated by thin horizontal lines and described in Sec. 2.4.

during the southern-hemispheric winter (CBH 173 m lower in June, July, August than during all other months, significant at the 95 % confidence level: independent t test). It is likely that during this time, FLC touches the ground even at the low-lying coastal stations (located on average ≈ 40m above sea level) frequently, leading to the observed agreement between ground fog and satellite-based FLC during this time (Fig. 3 a)). Between August and March, cloud-base height is significantly higher on

average and displays a higher variability, more frequently leading to situations where clouds are disconnected from the surface at the coast, but still might touch the ground further inland, leading to fog occurrences at stations located there (locations on average ≈ 490m above sea level).

## 3.2 Diurnal cycle of fog and low clouds

Based on the diurnally-stable FLC detection by Andersen and Cermak (2018), spatial information on the statistical properties

of the diurnal cycle of FLC can be analyzed. Figure 4 a) shows the average time of day when the FLC diurnal cycle typically starts. The start of the diurnal cycle is defined here as the first occasion after the diurnal FLC minimum during noon, when the relative FLC occurrence frequency reaches 10 % of the total range of its diurnal cycle at this location and is derived from 14 years of SEVIRI observations. To focus on the regions where FLC frequently occurs, pixels are only considered if they are located within 100 km to the coastline and feature a relative frequency of FLC occurrence of at least 5 %. It is apparent

from Fig. 4 a) that the start of the diurnal FLC cycle features distinct spatial patterns that are closely related to the distance



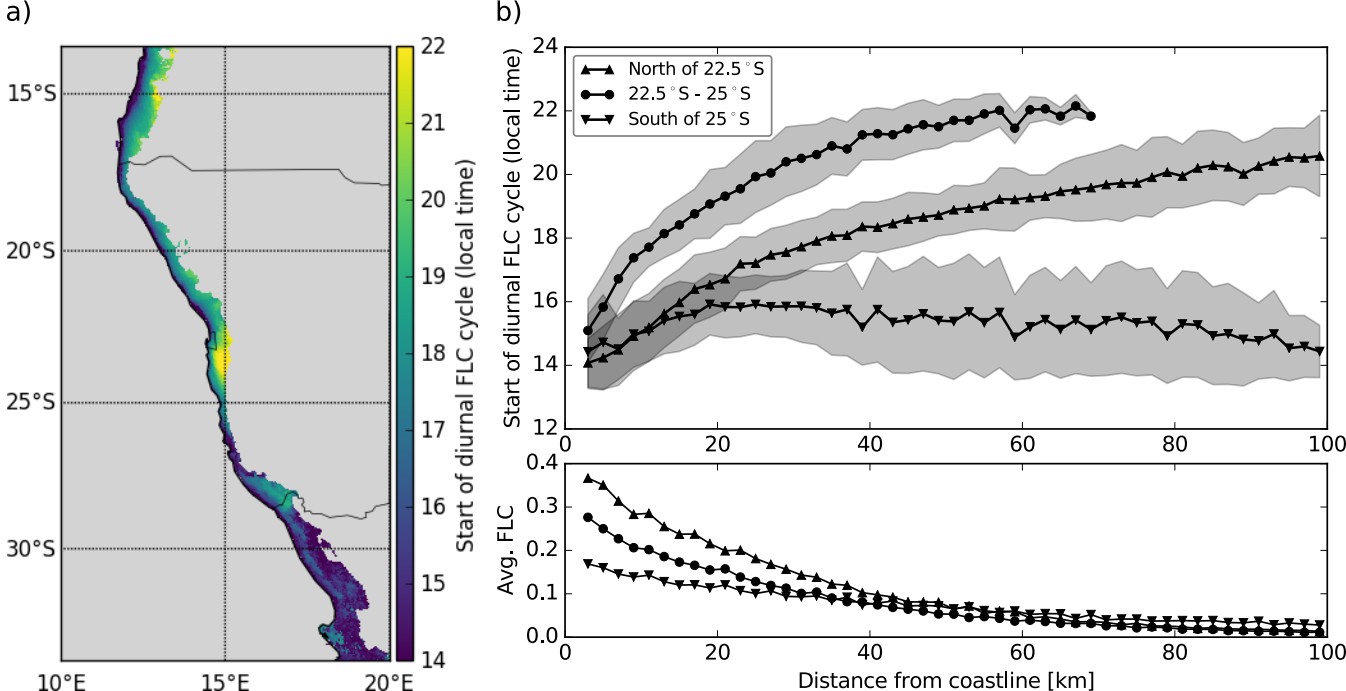

**Figure 4.** a) The time of the start of the diurnal FLC cycle on pixel level. Pixels are not considered which either are more than 100 km removed from the coastline or that feature a relative frequency of flc occurrence of less than 5 %. Shaded area illustrates mean +/- one standard deviation. b) Upper panel: The average timing of start of the diurnal FLC cycle as a function of average distance to the coastline. Lower panel: Average relative FLC occurrence frequency in the three subregions. The same pixels are considered as in panel a) and averaged in 2 km distance bins (x axis).

from the coastline, at least north of 25°S (r = 0.86 between 22.5°S and 25°S and r = 0.85 north of 22.5°S). As Andersen and Cermak (2018) argue, this is a clear indication of a region dominated by advective processes rather than radiation fog, contrasting findings from Kaseke et al. (2017). It should be noted that while the results are of statistical nature and thus reflect the dominant patterns, incidences of radiation fog are also likely to occur, at least in some locations. The apparent discrepancy

5    between these findings might be related to the limited sampling of the isotope analyses or due to a mixing of water from marine and continental sources as water vapor from local sources is additionally condensed at the front of the advected cold marine stratus.

More distinct spatial characteristics in the start time of the diurnal FLC cycle can be identified, as in the between 22.5°S and 25°S (circles in Fig. 4 b)), FLC typically starts to occur more than two hours later than in other regions along the southwestern

10    African coastline. The differences in timing between the three subregions are highly significant (significant at the 99 % confidence level, two-sided t test). South of 25°S, the diurnal cycle of FLC seems to start earlier and to only depend on the distance to the coastline up to a distance of ≈ 20 km (r = 0.42) and seems decoupled from the coast further inland (r = -0.20). The region at Alexander Bay seems to be an exception, where the diurnal cycle of FLC is similar to that of the northern regions. This may




be seen as a suggestion of subregional differences in the mechanisms leading to FLC formation. The lower panel of Fig. 4 b) shows the average FLC occurrence frequency in the three subregions as a function of the distance to the coastline that features a strong relationship, especially north of 25°S.

Figure 5 (upper panel) shows the time of the start of the diurnal FLC cycle between 22.5°S and 25°S in two different
5   time periods with contrasting vertical FLC characteristics. During the season of systematically higher-level FLC (September–November: high-FLC season), a distinct relationship between distance from coastline and the timing of FLC occurrence is apparent up to about 60 km inland. During the time of lower-level FLC (April, May, June: low-FLC season), this relationship is only apparent within ≈ 30 km of the coastline. It should be noted that the overall FLC occurrence frequency is also dependent on the distance from coastline (lower panel), and in inland regions, where no relationship between distance from coastline and
10   time of FLC occurrence is apparent, FLC occurrence is below 5 %. In these regions, assessments of the statistics of the diurnal cycle are limited by the overall accuracy of the detection algorithm (97 % - (Andersen and Cermak, 2018)), and the statistics of the diurnal cycle may be more susceptible to the influence of random misclassifications. In general, the slope of the relationship illustrated in the upper panel of Fig. 5 can be affected by the average advection speed, the fraction of advective FLC, and the partial contribution of random misclassifications.

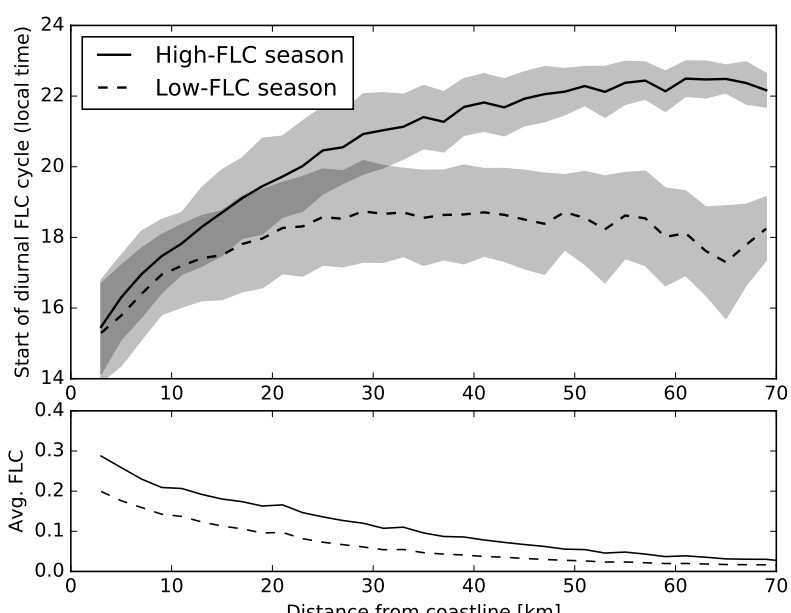

**Figure 5.** Upper panel: Start of diurnal FLC cycle in the central Namib as a function of distance from the coastline. Two different seasons are shown: high-FLC season (September, October, November) and low-FLC season (April, May, June). Shaded area illustrates mean +/- one standard deviation. Lower panel: Average relative FLC occurrence frequency in the two seasons. The same pixels are considered as in the upper panel and averaged in 2 km distance bins (x axis).





## 4 Conclusions and outlook

In this study, Namib-region fog and low-cloud patterns are analysed based on data from multiple satellite sensors as well as station measurements.

The seasonal cycle of satellite-derived FLC occurrence is found to have a distinct latitudinal dependence. In the Angolan
regions north of ≈17.5°S, FLC occurrence peaks between July and October, whereas in Namibia between 20°S and 25°S, FLC occurs mostly between August and February. This pattern may be explained by a seasonal shift in the dynamic conditions that lead to the inland-advection of marine low clouds. On seasonal scales, the spatiotemporal FLC occurrence indicates a connection to the southeastern Atlantic stratocumulus cloud deck. As such, process knowledge from studies on the heavily investigated stratocumulus clouds in this region (e.g., Adebiyi and Zuidema, 2018; Andersen and Cermak, 2015; Diamond
et al., 2018; Fuchs et al., 2017, 2018; Gordon et al., 2018; Painemal et al., 2014; Yuter et al., 2018) may be applicable to Namib-region FLC, and vice versa.

Satellite-derived seasonal patterns of FLC are compared to ground-based measurements of fog occurrence from the FogNet stations in the central Namib. While the seasonal patterns agree qualitatively and quantitatively for inland stations, they feature contrasting patterns at coastal stations. This can likely be explained by seasonal patterns in cloud-base altitude that determines
whether a low-level cloud touches the ground (fog) or not. Observations from CALIPSO and SCIAMACHY suggest that on average, clouds in coastal regions seem to be disconnected from the surface more frequently between August and February, where the satellite observations strongly overestimate station measured ground-fog occurrence.

Coherent spatial patterns of the diurnal cycle of FLC occurrence in the Namib could be observed for the first time using the algorithm developed by Andersen and Cermak (2018). Generally, the timing of FLC occurrence seems to be tightly connected
to the proximity of the coastline, where the diurnal cycle of FLC starts systematically earlier at the coast than further inland. This is a strong indication for a dominant role of advection for the climatological patterns of FLC in the region, contrasting the interpretation of findings from isotope analyses by Kaseke et al. (2017). In the central Namib, the diurnal cycle of FLC is found to start more than 2 hours later than in most regions along the coastline. This may be caused by local advection patterns of FLC. The key findings regarding the seasonal and diurnal patterns of FLC are summarized schematically in Fig. 6 and lead to a
more complete view on Namib-region FLC. The results of this study highlight the advantages of combining ground and space-based (active and passive sensoric) measurements. Future research should focus on a further characterization of the dynamical conditions and drivers that determine diurnal and seasonal variability and vertical structure of FLC, as well as processes that influence its diurnal cycle.





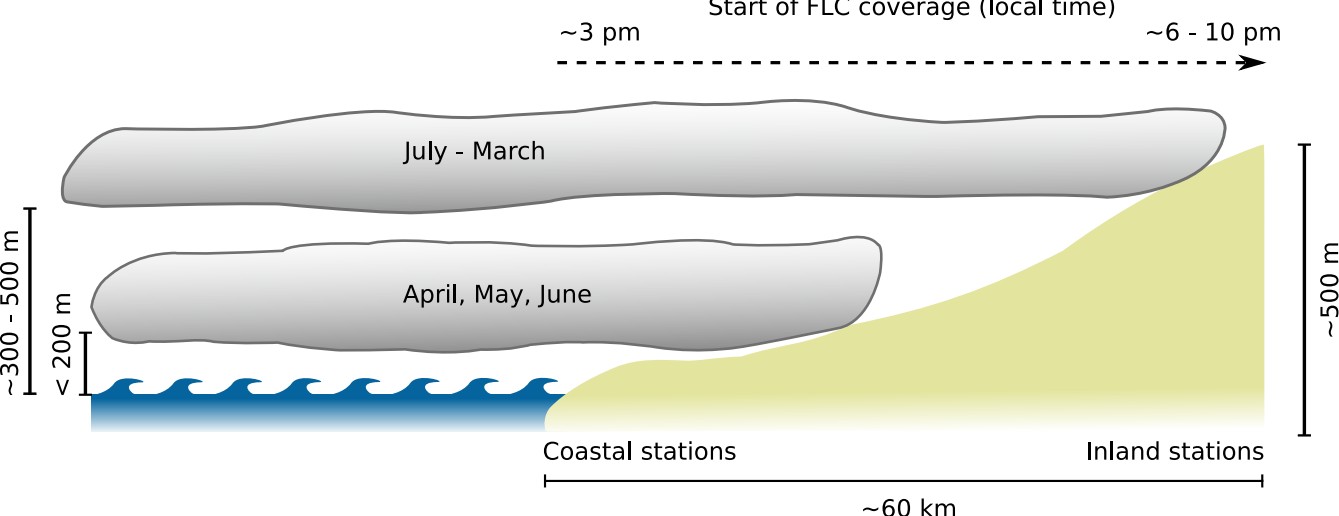

**Figure 6.** Schematic illustrating the observed seasonal cycle in cloud vertical characteristics and the dependence of the diurnal FLC cycle on the distance to the coastline.

## Appendix A: Abbreviations of FogNet stations

Aussinanis: AU

Coastal Met: CM

Conception Water: CW

5   Garnet Koppie: GK

Gobabeb Met: GB

Kleinberg: KB

Marble Koppie: MK

Saltworks: SW

10   Sophies Hoogte: SH

Station 8: S8

Vogelfederberg: VF

*Code and data availability.* Code and data are available





*Author contributions.* HA and JC had the idea for the analysis, HA obtained and analyzed most of the data sets, conducted the original research and wrote the manuscript. IS and LL contributed to data analysis, and RV provided the quality controlled FogNet data. JC, IS, LL and RV contributed manuscript preparation and the interpretation of findings.

*Competing interests.* The authors declare that they have no conflict of interest.

5   *Acknowledgements.* Funding for this study was provided by Deutsche Forschungsgemeinschaft (DFG) in the project Namib Fog Life Cycle Analysis (NaFoLiCA), CE 163/7-1. L. Lelli has been financially supported by the European Space Agency (ESA) via the Living Planet Fellowship for the STARCLINT (STatistics of AeRosol and CLoud INTeractions) project and by the German Science Foundation (DFG) in the framework of the Transregional Collaborative Project TR 172 AC3 (ArctiC Amplification: Climate relevant Atmospheric and surfaCe processes and feedback mechanisms). The authors would like to thank the Gobabeb Research and Training Centre for access to the sta-
10   tion measurements and gratefully acknowledge the Gobabeb maintenance team for their efforts in the field. We thank Mary Seely for her contributions in the development of FogNet.



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
