# Peer review of "Spatiotemporal dynamics of fog and low clouds in the Namib unveiled with ground and space-based observations"

_Atmospheric Chemistry and Physics, 2018_

## Referee Comment (RC1) · Olivier (Referee) · 15 Jan 2019

General comments: While fog and low cloud (FLC) form the lifeblood of desert flora and fauna in the Namib, their occurrence are considered to be hazardous to human activities such as aviation and shipping. It is thus important to understand where and when FLC occur. This paper examines the spatial and temporal incidence of FLC in the Namib, with special reference to the Central Namib. It also aims to help understand the processes driving the occurrence of FLC. Both ground based data and a variety of geostationary satellite based observations such as SEVIRI, CALIPSO, SCIAMACHY are used for this purpose. The use of these space-based observation adds a novel

aspect to research. The two guiding hypotheses were successfully addressed and found to be valid. The paper is well-written and a pleasure to read. It fulfils all the criteria required for publication in a high-impact journal.

Specific comments: Of special importance is the simple and clear explanation given for the anomaly between the ground- based and satellite based observations of the seasonal incidence of FLC in coastal regions. Unfortunately, this implies that satellite-based data cannot be used to examine the extent of fog over the coastal and adjacent maritime regions. The final recommendation by the authors i.e. that 'future research should focus on further characterization of the dynamical conditions and drivers that determine diurnal and seasonal variability and vertical structure of FLC is extremely important'. This should include the seasonal shift in location and intensity of the S. Atlantic and sub continental high pressure systems over southern Africa and their impact on the height of the inversion layer over the Namib. This together with the influence of the Namib-Benguela Upwelling System will provide a comprehensive picture and explanation of surface fog occurrence in the coastal regions.

Suggestions: Use colours for b in figure 4 rather than triangles. It will facilitate the interpretation of the results.

Please note: Research was conducted on fog in the Namib by Olivier J 1992: Some spatial and temporal aspects of fog in the Namib. SA Geograaf, 19(1/2) 106 - 126. If required, I can send a copy of the article to the authors.

Technical corrections: p2, 26: replace 'nearby' with 'near' p3, 9: is CALIPSO level '2 5 km' correct? p5, 27: word missing after 'over...,' p10, 22: ..In the central Namib, the diurnal cycle... are you referring to the whole central Namib or to the coastal region in the central Namib?
* * *

---

## Referee Comment (RC2) · Westerhuis (Referee) · 25 Jan 2019

**General comments**

Andersen et al. present a study about the spatial and temporal patterns of fog and low clouds in the Namib. The present paper extends the knowledge gained from earlier studies via the combination of ground measurements (fog precipitation, relative humidity and cloud base height) with data from several satellite platforms (spatial extent, cloud base height and cloud top height). They investigate spatial, seasonal and temporal patterns. In the end, they derive a conceptual model for fog and low clouds in the Namib.

[Figure]

The main conclusions in this study are generally comprehensible and well substantiated by the results. I congratulate the authors for deriving the very nicely summarising schematic of the seasonal FLC cycle. My main point to improve the paper in the revisions is that the information conveyed to the reader could be written in a more easily understandable and more concise way. Especially at the beginning, it was not obvious to me which phenomenon was referred to with "satellite observations differ from station measurements" as comparing ground fog measurements with satellite fog and low clouds observations obviously only tells half of the story.

The figures are nicely drafted and I only made a few suggestions to add small features which could facilitate it for the reader to grasp the content (see specific comments).

The text is carefully written, some details to improve are pointed out in the technical corrections.

Overall, the paper is understandable and interesting and I recommend publication after minor revisions.

**Specific comments**

0) Abstract

- P1L4-6: The sentence "...observed seasonal patterns derived from satellite observations differ from station measurements..." is misleading, it should be clarified that station measurements only observe ground fog.

1) Introduction

- P2L3-4: Again, it should be stated more clearly what kind of station measurements are compared to satellite data.

- P2L5: Explain better what you mean with "seasonal cycles of formation mechanisms".

2) Introduction

- I see a benefit in adding a small table or graph summarising the used datasets including availability (time period) and resolution (time and space).

- Section 2.3 is more difficult to read than the ones before. Shorter, less nested sentences could improve readability.

3) Results and Discussion

- Figure 4: I suggest to indicate the three separated regions from b) also on the map in a). And to me it is not obvious which data are comprised in one circle/triangle.

- The text could be somewhat sharpened: Eg P7L15: What do you mean with "distinct spatial patterns"?; P9L1: Which are the "subregionally different mechanisms"?; P9L3: Can you elaborate the relationship you are referring to in "FLC occurrence frequency...features a strong relationship"? → These sentences sound complicated but do not provide much information to the reader. My suggestion is to either delete them or explain more specific what you want the reader to know.

- P9L8: How do you interpret this discrepancy between the high- and low-level FLS season? Can you indicate the distance where FLS occurrence is below 5% in Fig. 5?

4) Conclusions and outlook

- P10L17: Do you want to say that satellite observations really "overestimate ground fog" or that based on these observations it is just not possible to distinguish between fog at the ground and low clouds lifted from the surface?

**Technical corrections**

- Overall: The term FLC is used inconsistently. Either use plural or singular and always use the abbreviation after it is introduced (eg P2L16+17).

- P1L8: This should be "25° S", not "25° N" I presume.

- P1L9 and P8L1: Please explain "r".

- P2L1: patterns "of" fog

- P2L25: In Fig. 1a) the western boundary is 10°E. For consistency reasons, I suggest taking the same extent as in Fig. 2a).

- P3L9: Although correct, a reader who is not familiar with CALIPSO products might think that "level 2 5 km" is a typo. The sentence could be rearranged.

- P3L11: To my knowledge, dates should be written in the form "June 13, 2006".

- P4L1 and L19: Indicate size also in km, for easier comparison with SEVIRI data.

- P5 title: Suggestion: Fog and low cloud "spatial" patterns

- P5L27: unfinished sentence

- P8 figure caption: "fls" should be in capitals.

- P8L8: Omit the "the" at the end of the line.

Stephanie Westerhuis

---

## Author Comment (AC1) · 31 Jan 2019

**Spatiotemporal dynamics of fog and low clouds in the Namib unveiled with ground and space-based observations**

**— EDITOR AND REVIEWER RESPONSES —**

Hendrik Andersen, Jan Cermak, Irina Solodovnik, Luca Lelli and Roland Vogt contact: hendrik.andersen@kit.edu

We would like to thank the co-editor Dr. Frank Eckardt and the two reviewers Dr. Jana Olivier and Stephanie Westerhuis for their careful reviews of the manuscript and their constructive criticism. Comments by the co-editor/referees are colored in blue, our replies or comments are colored in black.

**Response to the Co-Editor**

This is a very interesting paper that provides a first insight into the behaviour of fog fusing satellite and ground observations.

I have two comments

One detailed and one general.

Detailed comment.

Figures 2,3 and 4. These are a bit cryptic given the use of acronyms which need to be retrieved one by one from the text. I would encourage spelling these out in the captions. Furthermore, the linkages between the series of figures are not great.

Figure 2b) please show the pixels that have been used to derive 2c) Please extend the latitudes from a and b into c.

Thank you for the detailed comments on the figures. We agree that the mentioned aspects of the figures can be improved upon. The newly produced version of figure 2 is shown below (Fig. 1) and included in the revised manuscript.

[Figure]

Figure 1: A satellite-based climatology of relative fog and low cloud occurrence frequency derived by using the algorithms presented in Cermak (2018) (a)) and Andersen and Cermak (2018) (b)), based on the nearly complete data records of CALIPSO (2006–2017) and SEVIRI (2004–2017). The seasonality (c)) is computed by averaging pixels from (b) in coastal regions (maximum 100 km distance to coastline) with frequent FLC occurrence (minimum of 5 % relative FLC occurrence in the 14-year climatology shown in b)). The regions used for averaging in c) lie within the orange contours in b).

Figure 3) spell out CTB and CTH

Also, the fact that CL31 is at CM needs to be extracted from the main text. This is very confusing. Why is there a change in CM and CL31 for July and August? Why is there no line for the CL31 observations? Also, what is ASL and AGL?

 We have incorporated the suggestions into the figure and agree that this improves its clarity. ASL and AGL stand for above sea level and above ground level, respectively. This is now written out in the caption.

[Figure]

Figure 2: c) Medians, 25th and 75th percentiles of monthly averaged CBH and CTH in the central Namib based on SCIAMACHY (above ground level; 22.5°S-24.0°S and 14.25°E–15.5°E, 2003–2009) and CALIPSO (above sea level; 22.5°S-24.0°S and 14.0°E–15.5°E, 2006–2017) observations, respectively. Ceilometer CBH observations (above sea level) are only available since September 2017. Ceilometer positions (CoastalMet from September–June and Swakopmund July and August) and sensitivity limits are illustrated by thin horizontal lines and described in Sec. 2.4.

Figure 4) please depict the areas used to make in 4b in 4a) as boxes or state the northern and southernmost extent of these observations.

This is a good suggestion. We have now incorporated lines to illustrate the southern/northern boundaries of the three regions and included markers in a) to visually link the panels. The result is shown below (Fig. 3) and is included in the revised version of the manuscript.

[Figure]

Figure 3: a) The time of the start of the diurnal FLC cycle on pixel level. Pixels are not considered which either are more than 100 km removed from the coastline or that feature a relative frequency of FLC occurrence of less than 5 %. The dashed horizontal lines indicate the northern/southern boundaries of the three regions considered in b), with markers illustrating their respective association. b) Upper panel: The average timing of start of the diurnal FLC cycle as a function of average distance to the coastline. Shaded area illustrates mean +/- one standard deviation. Lower panel: Average relative FLC occurrence frequency in the three subregions. The same pixels are considered as in panel a) and are averaged in 2 km distance bins (x axis).

The appendix now provides a full list of all acronyms used in the manuscript.

On a more general note, the paper is very descriptive and not explanatory.
It would be great to tie these observations into our understanding of regional
Synoptics and local winds. The work by Tyson would be particularly apt to
consider. At the moment there are linkages to processes even at the most basic
level. If this is to happen elsewhere at least a brief description and explanation
would be welcome.

Thank you for this comment. We agree that the main focus of the manuscript is to characterize the spatiotemporal patterns of FLC in the region, with some limited inferences of processes. We agree that more research is needed to understand the role of synoptic scale and local drivers, and are currently investigating these aspects within the NaFoLiCA research project. We do feel that these aspects are not within the scope of the current manuscript, though, as this topic is complex and demands a thorough treatment. We do now state our plans to tackle these research questions more clearly in the last paragraph of the revised manuscript: *The interplay of large-scale dynamics with local winds (Tyson and Seely, 1980; Olivier, 1992, and sources therein), (sea) surface characteristics (Olivier, 1995), radiative transfer and aerosols is likely to explain fog and low cloud occurrence and variability in the Namib desert. The exact manner, however, by which the various processes determine this complex system and its observed spatiotemporal dynamics is still unclear. Future research is thus needed to more fully understand the processes that lead to the variability in spatial patterns, overall coverage, vertical structure and life cycle of FLC, as well its capacity to serve as a water source for ecosystems. Within the ongoing research project Namib Fog Life Cycle Analysis (NaFoLiCA), these aspects will be studied using a combination of satellite data, ground-based measurements and numerical models.*

**Response to Dr. Jana Olivier**

General comments: While fog and low cloud (FLC) form the lifeblood of desert flora and fauna in the Namib, their occurrence are considered to be hazardous to human activities such as aviation and shipping. It is thus important to understand where and when FLC occur. This paper examines the spatial and temporal incidence of FLC in the Namib, with special reference to the Central Namib. It also aims to help understand the processes driving the occurrence of FLC. Both ground based data and a variety of geostationary satellite based observations such as SEVIRI, CALIPSO, SCIAMACHY are used for this purpose. The use of these space-based observation adds a novel aspect to research. The two guiding hypotheses were successfully addressed and found to be valid. The paper is well-written and a pleasure to read. It fulfils all the criteria required for publication in a high-impact journal.

Thank you for reviewing the manuscript and for the positive feedback.

Specific comments: Of special importance is the simple and clear explanation given for the anomaly between the ground- based and satellite based observations of the seasonal incidence of FLC in coastal regions. Unfortunately, this implies that satellite-based data cannot be used to examine the extent of fog over the coastal and adjacent maritime regions. The final recommendation by the authors i.e. that 'future research should focus on further characterization of the dynamical conditions and drivers that determine diurnal and seasonal variability and vertical structure of FLC is extremely important'. This should include the seasonal shift in location and intensity of the S. Atlantic and sub continental high pressure systems over southern Africa and their impact on the height of the inversion layer over the Namib. This together with the influence of the Namib-Benguela Upwelling System will provide a comprehensive picture and explanation of surface fog occurrence in the coastal regions.

Thank you for this comment. We agree wholeheartedly that the aspects mentioned by Dr. Jana Olivier are highly relevant and could significantly expand our current system understanding. We are in the process of investigating the role of large scale dynamics and SST for FLC occurrence patterns on different time scales. However, we feel that this is not within the scope of the current manuscript. As mentoined above, we now describe future goals more clearly in the revised version of the manuscript.

Suggestions: Use colours for b in figure 4 rather than triangles. It will facilitate the interpretation of the results.

We agree that the new version of the figure (Fig. 3 in this document) is easier to interpret due to the added coloring.

Please note: Research was conducted on fog in the Namib by Olivier J 1992: Some spatial and temporal aspects of fog in the Namib. SA Geograaf, 19(1/2) 106 - 126. If required, I can send a copy of the article to the authors.

Thank you for the reference, this was an oversight on our part. We have been able to locate the article and it is now properly cited in the manuscript.

Technical corrections: p2, 26: replace 'nearby' with 'near'

We have now corrected this in the manuscript.

p3, 9: is CALIPSO level '2 5 km' correct?

Yes, this is correct.

p5, 27: word missing after 'over...,'

Yes, this is now corrected in the revised manuscript.

p10, 22: ..In the central Namib, the diurnal cycle... are you referring to the whole central Namib or to the coastal region in the central Namib?

This refers to the "whole" central Namib as defined in the manuscript. Basically, this is the "yellow blob" in Fig. 4a), where FLC occurs systematically later than in the adjacent regions to the north and south.

**Response to Stephanie Westerhuis**

**General comments**

Andersen et al. present a study about the spatial and temporal patterns of fog and low clouds in the Namib. The present paper extends the knowledge gained from earlier studies via the combination of ground measurements (fog precipitation, relative humidity and cloud base height) with data from several satellite platforms (spatial extent, cloud base height and cloud top height). They investigate spatial, seasonal and temporal patterns. In the end, they derive a conceptual model for fog and low clouds in the Namib.

The main conclusions in this study are generally comprehensible and well substantiated by the results. I congratulate the authors for deriving the very nicely summarising schematic of the seasonal FLC cycle. My main point to improve the paper in the revisions is that the information conveyed to the reader could be written in a more easily understandable and more concise way. Especially at the beginning, it was not obvious to me which phenomenon was referred to with "satellite observations differ from station measurements" as comparing ground fog measurements with satellite fog and low clouds observations obviously only tells half of the story.

The figures are nicely drafted and I only made a few suggestions to add small features which could facilitate it for the reader to grasp the content (see specific comments).

The text is carefully written, some details to improve are pointed out in the technical corrections.

Overall, the paper is understandable and interesting and I recommend publication after minor revisions.

Thank you for reviewing the manuscript and for the positive feedback.

**Specific comments**

P1L4-6: The sentence "...observed seasonal patterns derived from satellite observations differ from station measurements..." is misleading, it should be clarified that station measurements only observe ground fog.

This is now clarified in the revised version of the manuscript.

P2L3-4: Again, it should be stated more clearly what kind of station measurements are compared to satellite data.

This is now clarified in the revised version of the manuscript.

P2L5: Explain better what you mean with "seasonal cycles of formation mechanisms".

The text now states: "This could be related to seasonally varying mechanisms responsible for fog formation/type or due to a seasonal cycle in vertical characteristics of FLC in this region,[...]"

I see a benefit in adding a small table or graph summarising the used datasets including availability (time period) and resolution (time and space).

Thank you for this comment. We feel that an additional table would introduce quite a bit of redundancy to the manuscript and would thus prefer to keep the data descriptions in their current state.

Section 2.3 is more difficult to read than the ones before. Shorter, less nested sentences could improve readability.

We have rephrased some sentences in this section for clarity.

Figure 4: I suggest to indicate the three separated regions from b) also on the map in a). And to me it is not obvious which data are comprised in one circle/triangle.

For added clarity, we now show region boundaries and markers for b) in a). (Fig. 3 in this document).

The text could be somewhat sharpened: Eg P7L15: What do you mean with "distinct spatial patterns"?

Yes, this was not clearly written. The sentence now reads: "It is apparent from Fig. 4 a) that the start of the diurnal FLC cycle is closely related to the distance from the coastline, at least north of 25°S (r = 0.86 between 22.5°S and 25°S and r = 0.85 north of 22.5°S)."

P9L1: Which are the "subregionally different mechanisms"?

The close relationship between the start of the diurnal FLC cycle and the distance from the coastline suggests dominant advective processes north of 25°S. South of 25°S, this is no longer apparent. This leads us to the conclusion that advective mechnasms are unlikely to dominate in this region, however, as of now there are no observational clues to what extent specific mechanisms contribute to the formation of FLC in the southern region.

P9L3: Can you elaborate the relationship you are referring to in "FLC occurrence frequency...features a strong relationship"? → These sentences sound complicated but do not provide much information to the reader. My suggestion is to either delete them or explain more specific what you want the reader to know.

In the revised version of the manuscript this is now more clearly described: "The lower panel of Fig. 4 b) shows the average FLC occurrence frequency in the three subregions as a function of the distance to the coastline that features a strong relationship, especially north of 25°S. While this is a typical feature of coastal fog (e.g., Olivier, 1992), it serves as an additional indication that the region south of 25°S is not influenced by marine airmasses to the same extent as regions further north."

P9L8: How do you interpret this discrepancy between the high- and low-level FLS season? Can you indicate the distance where FLS occurrence is below 5% in Fig. 5?

Based on the results it is hard to say what exatly is responsible for the observed seasonal differences. We do not want to speculate and thereby just state that *In general, the slope of the relationship illustrated in the upper panel*

*of Fig. 5 can be affected by the average advection speed, the fraction of advective*

*FLC, and the partial contribution of random misclassifications.* We do not see 5% as a strict threshold under which you cannot interpret the results any more. We rather state that lower FLC occurrence frequency also lowers the confidence in derived statistics, e.g., in those related to the diurnal cycle, due to the factors outlined by the sentence stated above.

P10L17: Do you want to say that satellite observations really "overestimate ground fog" or that based on these observations it is just not possible to distinguish between fog at the ground and low clouds lifted from the surface?

We argue that the probability of satellite-derived FLC being ground fog shifts with season and location. Using FLC for an estimate on fog occurrence at coastal locations between August and February would be specifically prone to an overestimation of fog occurrence frequency.

**226 Technical corrections**

Overall: The term FLC is used inconsistently. Either use plural or singular and always use the abbreviation after it is introduced (eg P2L16+17).

The term FLC/FLCs is now used consistently in the updated version of the manuscript. In specifically relevant sentences of the manuscript, as e.g. the sentence pointed out here, we deliberately chose to write out fog and low clouds instead of using the abbreviation. This is intended to help readers who are just skimming over the paper to understand the most relevant sentences even though they might not know all of the abbreviations.

P1L8: This should be "25°S", not "25°N" I presume.

Yes, of course you are right. This is now corrected in the manuscript.

P1L9 and P8L1: Please explain "r".

This should be more clear in the current manuscript.

P2L1: patterns "of" fog

Yes, this is now corrected in the manuscript.

P2L25: In Fig. 1a) the western boundary is 10°E. For consistency reasons, I

suggest taking the same extent as in Fig. 2a).

The western extent of the figures was chosen deliberately. 10°E makes sense for Fig. 1a) and Fig. 2b), as no information content would be added by further extending the figure over the ocean. Fig. 2a) shows the spatial connection of the FLC field over the coast with the stratocumulus field in the southeastern

Atlantic. We would thus prefer to keep the figures at their current state.

P3L9: Although correct, a reader who is not familiar with CALIPSO products might think that "level 2 5 km" is a typo. The sentence could be rearranged.

As this seems to be the official product name, we would like to keep the sentence in its current form.

P3L11: To my knowledge, dates should be written in the form "June 13, 2006".

Yes, indeed, we have corrected this in the revised version of the manuscript.

P4L1 and L19: Indicate size also in km, for easier comparison with SEVIRI

data.

This is technically not possible, as the size of a 1°x1° area depends on its latitude.

P5 title: Suggestion: Fog and low cloud "spatial" patterns

Yes, we agree that this is more accurate. We have changed the title accordingly.

P5L27: unfinished sentence

We have corrected the sentence.

P8 figure caption: "fls" should be in capitals.

Yes, this is now corrected in the manuscript.

P8L8: Omit the "the" at the end of the line.

We have corrected the sentence.

**References**

Andersen, H. and Cermak, J. (2018). First fully diurnal fog and low cloud satellite detection reveals life cycle in the Namib. *Atmospheric Measurement Techniques*, 11(July):5461–5470.

Cermak, J. (2018). Fog and low cloud frequency and properties from active-sensor satellite data. *Remote Sensing*, 10(8):1–7.

Olivier, J. (1992). Some spatial and temporal aspects of fog in the Namib. *South African Geographer*, 19(1-2):106–126.

Olivier, J. (1995). Spatial distribution of fog in the Namib. *Journal of Arid Environments*, 29(2):129–138.

Tyson, P. D. and Seely, M. K. (1980). Local winds over the central Namib. *South African Geographical Journal*, 62(2):135–150.

---

## Author Response (AR2)

**Spatiotemporal dynamics of fog and low clouds in the Namib unveiled with ground and space-based observations**

**— EDITOR RESPONSES —**

Hendrik Andersen, Jan Cermak, Irina Solodovnik, Luca Lelli and Roland Vogt

contact: hendrik.andersen@kit.edu

We thank the co-editor Dr. Frank Eckardt for his detailed comments on the manuscript. In the following response, comments by the co-editor are colored in blue, our replies or comments are colored in black.

**Response to the Co-Editor**

These comments are not technical but minor and are mainly concerned with clarification of figures etc.

1)Appendix A

Which of these stations are considered coastal and which are inland?

This attribution should also be clarified in Figure 3 a and b as well as Figure 1 b as well as section 2.4

Referring to colors alone might be sufficient but could again be spelled out.

A legend has been included in Fig. 1 b) and the mentioned text passages now state the attribution of the stations more clearly or better reference to the relevant Figure.

Figure 2 a and b, Are these annual averages without any diurnal bias? It might be useful to learn more about how this previous work, as cited, relates to this study. 2 c is an excellent figure.

Thank you for the feedback on Fig. 2. The caption now includes information on temporal sampling so that the reader does not have to look this up in Sec. 2.1 and 2.2, where the two referenced studies are described.

Figure 4) Upper and Lower Limit of Northern and Southern Boundary are still not clear. Does the dashed line represent the 22.5 degrees? If so please label it.

Having 2 circle symbols there is confusing.

Please clearly indicate the latitude from which each of the 3 lines was obtained.

Thank you for the valuable comments on Fig. 4. We have changed the way the panels are linked in a) and now state the northern and southern boundaries of the regions more clearly in the legend of b). We feel that this has markedly improved the clarity of the figure.

The interpretation in Section 4 does at no point take into account topography in relation to

1) River valleys

2) Escarpment v Coastal Plains

3) The surface roughness (i.e. dune fields)

The data here seems to suggest that these play a role in the fog distribution patterns as observed in Figure 4 and 5.

A paragraph to that effect might be useful.

We agree that these aspects play an important role for FLC patterns in the region, albeit partially on smaller scales. While our analysis is focused on the large-scale patterns and their seasonal and diurnal characteristics, we have now included these aspects in the discussion of the results on pages 6, 8, 9 and 11 of the updated version of the manuscript.

Figure 1 could be improved in this regard too. The topography of the coastal plain plays an important role and the lack of integration and illustration via a DEM as in the Andersen and Cermak paper is an omission here. Places are mentioned without being depicted in the map. Where is Alexander Bay?

Yes, we agree that the topography along the coastline plays an important role for fog and low clouds in the Namib. So far, the topography was only considered for the central Namib (isolines in Fig. 1 b)). Now, an isoline at 1000 m above

sea level is included in Fig. 1 a) to approximate the Great Escarpment and the eastern boundary of the Namib desert. For a better reference, Walvis Bay and Alexander Bay are now pointed out in the map as well.

Minor points

Page 2 line 5 – formation/type is ambiguous

This is now corrected in the current version of the manuscript.

Page 2 line 9 Uncertainties also exist related to the ... is clumsy phrasing and wording

The sentence has been rephrased for clarity.

Page 3 line 7 it has shown good skill... is clumsy phrasing and wording

The sentence is modified in the updated version of the manuscript.

Page 6 line 6 However, in accordance to the comparison.... is clumsy phrasing and wording

The sentence has been modified for clarity.

Page 8 line 5 perhaps insert the "here " after results, referring to this study

Yes, this is now corrected in the current version of the manuscript.

Non-public comments to the Author:

Overall this is a very important paper with a wealth of analyses at its core. Deciphering it takes a bit of time since figures appear small and a little cryptic. I find the spatial handling of the data and observed patterns superficial and rushed. There is also little integration and consideration for synoptic and wind

patterns. As such the paper remains quiet descriptive rather than explanatory. Thank you for the detailed comments and constructive criticism on the manuscript. The aim of this paper is to combine ground-based and space-based observations of fog and low clouds in the Namib and use these to characterize it in 4-dimensions. We are currently working on a follow-up manuscript that will focus, and lay out in much greater detail, the (thermo)dynamical patterns and mechanisms that are associated with FLC occurrence.

[revised manuscript text omitted]